# Extra Virgin Oil Polyphenols Improve the Protective Effects of Hydroxytyrosol in an In Vitro Model of Hypoxia-Reoxygenation of Rat Brain

**DOI:** 10.3390/brainsci11091133

**Published:** 2021-08-26

**Authors:** José Pedro De La Cruz Cortés, Inmaculada Pérez de Algaba, Esther Martín-Aurioles, María Monsalud Arrebola, Laura Ortega-Hombrados, María Dolores Rodríguez-Pérez, María África Fernández-Prior, Alejandra Bermúdez-Oria, Cristina Verdugo, José Antonio González-Correa

**Affiliations:** 1Departmento de Farmacología, Facultad de Medicina, Instituto de Investigación Biomédica (IBIMA), Universidad de Málaga, 29010 Málaga, Spain; lauraortegah@outlook.es (L.O.-H.); mariaperezrodriguez@ymail.com (M.D.R.-P.); cristinaverdugocabello@gmail.com (C.V.); correa@uma.es (J.A.G.-C.); 2Clinical Laboratory Department, Hospital Público de Montilla, 14550 Córdoba, Spain; iperezdealgaba@gmail.com; 3UGC La Roca, Distrito Sanitario Málaga-Guadalhorce, 29001 Málaga, Spain; esther.uma@hotmail.com; 4UGC Laboratorio Clínico, Hospital de la Axarquía, AGSEMA, 29740 Málaga, Spain; mariam.arrebola.sspa@juntadeandalucia.es; 5Instituto de la Grasa, Consejo Superior de Investigaciones Científicas (CSIC), Ctra. Utrera Km 1, Campus Universitario Pablo de Olavide, Edificio 46, 41013 Seville, Spain; mafprior@ig.csic.es (M.Á.F.-P.); aleberori@ig.csic.es (A.B.-O.)

**Keywords:** hydroxytyrosol, extra virgin olive oil, polyphenols, neuroprotection

## Abstract

Hydroxytyrosol (HT) is the component primarily responsible for the neuroprotective effect of extra virgin olive oil (EVOO). However, it is less effective on its own than the demonstrated neuroprotective effect of EVOO, and for this reason, it can be postulated that there is an interaction between several of the polyphenols of EVOO. The objective of the study was to assess the possible interaction of four EVOO polyphenols (HT, tyrosol, dihydroxyphenylglycol, and oleocanthal) in an experimental model of hypoxia-reoxygenation in rat brain slices. The lactate dehydrogenase (LDH) efflux, lipid peroxidation, and peroxynitrite production were determined as measures of cell death, oxidative stress, and nitrosative stress, respectively. First, the polyphenols were incubated with the brain slices in the same proportions that exist in EVOO, comparing their effects with those of HT. In all cases, the cytoprotective and antioxidant effects of the combination were greater than those of HT alone. Second, we calculated the concentration–effect curves for HT in the absence or presence of each polyphenol. Tyrosol did not significantly modify any of the variables inhibited by HT. Dihydroxyphenylglycol only increased the cytoprotective effect of HT at 10 µM, while it increased its antioxidant effect at 50 and 100 µM and its inhibitory effect on peroxynitrite formation at all the concentrations tested. Oleocanthal increased the cytoprotective and antioxidant effects of HT but did not modify its inhibitory effect on nitrosative stress. The results of this study show that the EVOO polyphenols DHPG and OLC increase the cytoprotective effect of HT in an experimental model of hypoxia-reoxygenation in rat brain slices, mainly due to a possibly synergistic effect on HT’s antioxidant action. These results could explain the greater neuroprotective effect of EVOO than of the polyphenols alone.

## 1. Introduction

The beneficial effect of the intake of extra virgin olive oil (EVOO) for the prevention of cardiovascular diseases has been widely demonstrated [1]. High adherence to a Mediterranean diet model supplemented with approximately 50 mL of EVOO per day has been shown to reduce the incidence of coronary ischemic attacks, peripheral artery disease, and ischemic stroke, in addition to preventing the onset of obesity and type 2 diabetes [1,2].

In an experimental study, the neuroprotective effect of the administration of doses of EVOO equivalent to those recommended for humans was demonstrated, both in normal rats and in an experimental model of type 1 diabetes [3]. Fundamentally, it has been shown that, in an in vitro model of hypoxia-reoxygenation in the brain slices of rats, chronic treatment with EVOO results in a significantly lower rate of cell death [3,4].

Likewise, different chemical components of EVOO have been studied separately, and the cardioprotective and neuroprotective effects of EVOO have been attributed to the polyphenol most abundant in its composition, hydroxytyrosol (HT), [5]. These effects are explained by the antioxidant effect demonstrated for HT in various tissues and in experimental models of disease, such as diabetes, hypertension, and dyslipidemia [6,7].

When the different polyphenols were studied separately, none of them could fully explain the effect exerted by EVOO [8]. Specifically, in terms of its neuroprotective effect, HT is the compound that can best explain the effect of EVOO, but it does not do so entirely [8]. For this reason, taking into account the other polyphenols in the composition of EVOO [9,10], it could be postulated that there is an interaction between them when they reach nervous tissue and thus, together they exert the total observed effects of EVOO.

In this study, we investigated four of the main polyphenols present in EVOO: hydroxytyrosol (HT), tyrosol (Ty), 3′,4′-dihydroxyphenylglycol (DHPG), and oleocanthal (OLC). We chose these four EVOO compounds because they are recognized as simple polyphenols with more biological potency than others in EVOO [6]. Some others, such as oleuropein, act by way of their HT content; HT is released in the stomach when oleuropein is ingested orally. Hydroxytyrosol is considered to have the highest biological activity of all the polyphenols in EVOO [8]. The objective of this study was to assess the possible effects that Ty, DHPG, and OLC could exert on the in vitro cytoprotective, antioxidant, and antinitrosative effects of HT in an experimental model of hypoxia-reoxygenation in rat brain slices.

## 2. Materials and Methods

### 2.1. Materials

A thiobarbituric acid reactive substances colorimetric kit and 3-nitrotyrosine enzyme immunoassay kits were obtained from Cell Biolabs Inc. (Bionova Científica S.L., Madrid, Spain). A lactate dehydrogenase colorimetric kit (Cytotoxicity Detection Kit) was obtained from Roche Applied Science (Barcelona, Spain). All the other reagents, including the tyrosol and oleocanthal, were from Sigma Chemical Corp. (St. Louis, MO, USA).

Hydroxytyrosol was isolated by a hydrothermal treatment of the liquid phase obtained from alperujo (a by-product of the two-phase olive oil separation system) at 160 °C for 60 min [11]. The liquid was extracted by two-step chromatography fractionation. The final yield reached 99.6% purity relative to dry matter, using the process described by Fernández-Bolaños et al. [12]. The phenols were quantified using a Hewlett-Packard 1100 liquid chromatography system with an ultraviolet–visible detector. A Mediterranean Sea C18 analytical column (250 × 4.6 mm i.d.; particle size = 5 µm) (Teknokroma, Barcelona, Spain) was used at room temperature. The system was equipped with Rheodyne injection valves (20 μL loop). The mobile phases were 0.01% trichloroacetic acid in water and acetonitrile, with the following gradient during a total run time of 55 min: 95% initially, 75% at 30 min, 50% at 45 min, 0% at 47 min, 75% at 50 min, and 95% at 52 min until the run was complete. Quantification was carried out by peak integration at a 280 nm wavelength with reference to the calibrations obtained with external standards (Figure 1).

3′,4′-dihidroxifenilglicol, obtained from the olive oil by-product of the two-phase extraction system used in olive oil mills, was used for the isolation of the DHPG. The method for purifying DHPG was described and patented by Fernández-Bolaños et al. (2010) (WO2010070168A1). The method is based on physical chromatographic systems that allow for the extraction of natural compounds without any organic solvent or chemical or enzymatic reactions, realizing a purity over 95%, with reference to dry matter (Figure 1).

### 2.2. Study Design

The animals were two-month-old adult male Wistar rats (with body weights of 200–250 g). All the rats were used in accordance with the current Spanish legislation for animal care, use, and housing (EDL 2013/80847, BOE-A-2013-6271). The recommendations of the Guide for the Care and Use of Laboratory Animals (NIH publication No. 86–23, revised 1985) were followed, as was the Spanish Law on the Protection of Animals, where applicable. The study protocol was approved by the University of Malaga Ethics Committee for the Use of Animals (Ref. CEUMA31-2018-A) and the Consejería de Agricultura, Ganadería, Pesca y Desarrollo Sostenible, Junta de Andalucía (Department of Agriculture, Livestock, Fisheries and Sustainable Development of the Regional Government of Andalusia) (Ref. 9/07/2019/124).

### 2.3. Hypoxia-Reoxygenation Procedure

All the rats were anesthetized with pentobarbital sodium (40 mg/kg i.p.) and then decapitated with a guillotine. The brain tissue (except the cerebellum and brain stem) was cut transversally into 1 mm slices with a vibrating microtome (Campden Instruments, San Francisco, CA, USA). The slices were placed in a buffer (composition in mmol/L: 100 NaCl, 0.05 KCl, 24 NaHCO_3_, 0.55 KH_2_PO_4_, 0.005 CaCl_2_, 2 MgSO_4_, 9.8 glucose, pH 7.4) and perfused with a mixture of 95% O_2_ and 5% CO_2_ (Period A). After 30 min, the brain slices were placed in fresh buffer with no glucose and 3 mmol/L of CaCl_2_ and 0.001 mmol/L of MgSO_4_. A mixture of 95% N_2_ and 5% CO_2_ (hypoxia) was perfused for 20 min (Period B). Subsequently, the slices were placed in a buffer with glucose and perfused with a mixture of 95% O_2_ and 5% CO_2_ for 120 min (reoxygenation) (Period C).

The brain samples were analyzed before the period of hypoxia, after 20 min of hypoxic incubation, and after 120 min of reoxygenation. They were then frozen in liquid nitrogen and stored at −80 °C. All the analytical techniques were performed not more than 7 days after the samples were frozen.

### 2.4. Incubation of Polyphenol Compounds

Four of the main EVOO polyphenol compounds were tested. Hydroxytyrosol (HT) was considered the main compound, and tyrosol (Ty), 3′,4′-dihidroxifenilglicol (DHPG), and oleocanthal (OLC) were studied as possible modifiers of HT’s effects.

All the compounds were added from the beginning of the experiment (Period A) and kept until the end of the reoxygenation period (Period C). The concentrations of each compound were chosen according to the ranges of concentrations described for EVOO, with low, medium, and high polyphenol contents [13,14], as well as the effects demonstrated in some in vitro experiments in rat brain [8,15]. Table 1 shows the ranges of each compound according to the literature and the exact concentrations incubated in the experiments.

### 2.5. Analytical Techniques

All the techniques were carried out in a single-blind manner; that is, the persons who conducted the assays were unaware of the origin and nature of the samples.

#### 2.5.1. Lactate Dehydrogenase (LDH) Efflux

The LDH efflux in the incubation buffer was measured as an indirect representation of cell death. Enzyme activity was measured spectrophotometrically at 340 nm according to the manufacturer’s instructions. Briefly, in a 96-well microplate, 100 µL of the sample was incubated with 100 µL of a reaction solution containing iodotetrazolium chloride and sodium lactate. The incubation of the free buffer with the reaction solution was used as a blank. The plate was incubated for 30 min, and then, 50 µL of the stop solution was added. The optical absorbance at 490 nm and 600 nm was determined. The value obtained at 600 nm was subtracted from that obtained at 490 nm, and subsequently, the absorbance obtained for the blank was subtracted from all the values.

#### 2.5.2. Lipid Peroxidation

The brain tissue was homogenized in 50 mM phosphate-buffered saline with a pH of 7.0 (1/15 *w*/*v*). The resulting sample was centrifuged at 13,000× *g* for 15 min at 4 °C, the supernatant was separated, and aliquots were frozen at −80 °C until used for the measurement of lipid peroxidation.

Thiobarbituric acid reactive substances (TBARS) were measured as an index of lipid peroxides, whose main compound is malondialdehyde (MDA). Briefly, 100 µL of the samples or increasing concentrations of malondialdehyde were incubated at room temperature for 5 min. Subsequently, 250 µL of thiobarbituric acid (5.2 mg/mL, pH 3.5, buffered with sodium hydroxide) was added. All the samples were incubated at 95 °C for 45 min and centrifuged at 3000× *g* for 15 min, and the optical absorbance at 532 nm of the supernatant was determined in a 96-well plate. The equation of the malondialdehyde standard curve was calculated, and the data obtained for the brain samples were interpolated.

##### 2.5.3. 3-Nitrotyrosine

As an index of peroxynitrite formation, the concentration of 3-nitrotyrosine was measured. The brain slices were homogenized (1:10 wt/vol) in 100 mM KH_2_PO_4_/K_2_HPO_4_ and 0.1% digitonin (pH of 7.4). Then, they were centrifuged (5000× *g*, 10 min, 4 °C). The concentration of 3-nitrotyrosine in the supernatant was measured according to the manufacturer’s instructions for the enzyme immunoassay kit. Briefly, in a 96-well plate, 50 µL of sample and increasing concentrations of the 3-NTy standard or buffer (blank) were incubated with 50 µL of anti-3-NTy antibody for 1 h at room temperature. The plate was washed with wash buffer and incubated with 100 µL of secondary anti-3-NTy antibody for 1 h at room temperature. Subsequently, 100 µL of the substrate solution was added, and the optical absorbance at 450 nm was determined. The equation of the 3-NTy standard curve was calculated, and the data obtained from the brain samples were interpolated.

### 2.6. Statistical Analysis

The data in the text, tables, and figures are expressed as the mean ± standard error of the mean (SEM) for 6 animals. All the statistical analyses were performed with the Statistical Package for Social Sciences v25.0 (SPSS Co., Chicago, IL, USA). Unpaired Student’s *t*-tests were used to compare the differences between the means. In all cases, statistical significance was assumed at a value of *p* < 0.05.

## 3. Results

In the in vitro model of the hypoxia-reoxygenation of the brain slices of rats, an increase in LDH efflux (×4.6), lipid peroxidation (as the brain tissue TBARS concentration) (×7.3), and nitrosative stress (as the brain tissue 3-NTy concentration) (×9.0) occurred after the reoxygenation period, with respect to the preanoxia values (Table 2).

First, we carried out hypoxia-reoxygenation experiments in rat brain sections, incubating with the four polyphenol compounds according to their reported concentrations in EVOO, according to high, medium, or low EVOO polyphenol contents, and compared the results with those for HT alone at the same concentration. Figure 2 shows the results for LDH efflux, TBARS, and 3-nitrotyrosine, expressed as percentages for each variable with respect to the values obtained for the control samples after the reoxygenation period. The cytoprotective and antioxidant effects of the mixture of HT, Ty, DHPG, and OLC were significantly greater than those of HT alone, by 30–50%, except for 3-NTy, which was inhibited to the same extent by HT as by the mixture of polyphenols.

The four polyphenols showed cytoprotective and antioxidant effects and inhibited peroxynitrite production in a concentration-dependent manner (Figure 3). Table 3 shows the concentrations of the compounds that produced 50% inhibition of each variable after the reoxygenation period (IC_50_). Regarding the cytoprotective and antioxidant effects (LDH efflux and TBARS, respectively), HT and OLC showed significantly greater effects (IC_50_ in the 10^−5^ M range) than Ty and DHPG (IC_50_ in the 10^−4^ M range) did. The effects on the production of peroxynitrite (3-NTy) were similar in all cases (IC_50_ in the range 10^−5^ M).

When increasing concentrations of HT were used in the presence of Ty, the antioxidant and peroxynitrite-production-inhibitory effects were not quantitatively modified, but the cytoprotective effect of HT was reduced (Figure 4 and Table 3).

The concentration–effect curves for HT’s cytoprotective effect were shifted to the left (higher power) when 10 µM DHPG was present (the IC_50_ value was 65.1% lower), as were those for the antioxidant effect when 50 and 100 µM DHPG was present (the IC_50_ values were 47.3% and 12.8% lower, respectively), and all three concentrations caused leftward shifts for the inhibition of peroxynitrite formation (the IC_50_ values were 56.9%, 54.3%, and 48.9% lower, respectively) (Figure 5 and Table 3).

The incubation of OLC together with increasing concentrations of HT increased the cytoprotective effect of HT (Figure 6 and Table 3) (IC_50_ 75%–80% lower than those for HT alone). The antioxidant effect of HT was also increased in the presence of OLC (IC_50_ 40%–60% lower than those for HT alone). The inhibitory effect of HT on peroxynitrite production was not modified by the presence of OLC.

## 4. Discussion

The results obtained in this study show that the main polyphenols of EVOO interact under in vitro conditions in rat brain tissue, mainly in terms of the cytoprotective effect and possibly through an interaction of antioxidant effects and inhibitory effects on the production of peroxynitrite, among other possible mechanisms.

We chose the in vitro hypoxia-reoxygenation model using rat brain slices because it has been shown that, in this model, there is cell death accompanied by an increase in oxidative and nitrosative stress [16,17]. This was fulfilled in the present study, as can be seen in Table 2.

It is known that EVOO polyphenols, mainly HT, exert cytoprotective effects in various tissues and experimental models [8,18,19,20]. These effects are due to various mechanisms of action, but it is accepted that the demonstrated antioxidant effects of these compounds are the main axes of their actions [21]. In the experimental model used in this study, the inhibition of lipid peroxidation by the four compounds was verified (Table 3 and Figure 3), presenting the following order regarding the potency of their antioxidant effects: HT > OLC ≥ DHPG > Ty. However, the order in relation to inhibitory effects on peroxynitrite formation was DHP G> HT ≥ Ty ≥ OLC.

First, we wanted to verify the general hypothesis raised in this study, which was that the cytoprotective effect of HT does not alone explain the cytoprotective effect of EVOO with regard to its content of other polyphenols [13,14]. To confirm this, we incubated the four polyphenols in the same proportions as found within EVOO—for low, medium, and high polyphenol contents—and compared their effects with those of HT alone with the same proportions. We were able to observe that, in all cases, HT showed lower cytoprotective and antioxidant effects than the combination of polyphenols did, except for the inhibition of the production of peroxynitrite, for which HT seemed to carry the highest specific weight (Figure 2). Therefore, it was confirmed that the combination of polyphenols explained the cytoprotective effect of EVOO better than HT alone did.

The four compounds showed cytoprotective effects separately in this experimental model, although HT and OLC showed greater potency than Ty and DHPG did. Using the same model, this effect has been previously described for HT [8,15,16,17] and Ty [22,23] at the same proportions as in this study. OLC has demonstrated a neuroprotective effect on neuron-like SH-SY5Y cells induced with H_2_O_2_ at proportions similar to those in our study and at similar concentrations [24]. Another study showed a neuroprotective effect of OLC in Alzheimer’s models [25]. We found no studies in the consulted literature that demonstrated a neuroprotective effect of DHPG in the chemical form that EVOO contains.

It is indisputable that the antioxidant effects play a fundamental role in the effects of the studied compounds, and there are other mechanisms that possibly participate, such as the anti-inflammatory effects [26]. As can be seen in Table 3, except for those of DHPG, there is a parallel between the IC_50_ values for the inhibition of lipid peroxidation and the formation of peroxynitrite and LDH efflux.

From a pharmacological point of view, DHPG and OLC increase the cytoprotective effect of HT, while Ty decreases it. Ty has been reported to exert a poor cytoprotective effect in vitro [22], while its ex vivo effect resembles that of HT [23]. One possible explanation is that, in vitro, Ty acts on the same mechanisms as HT does but with much less potency, which would make it a partial agonist, that is, a weak cytoprotective compound if used alone and an inhibitor of the effect of HT when both are present in brain tissue. In these cases, when high concentrations of an AgP are incubated together with an Ag, the former behaves as an antagonist. In the experiments of this study, it was shown that Ty tended to inhibit the effects of HT as an antioxidant and inhibitor of peroxynitrite production, especially at higher concentrations, which could partially support the hypothesis. Furthermore, it has been described that, under ex vivo conditions, some Ty is transformed into HT in the liver [27], which could also explain the greater effect in this type of experiment [23].

DHPG increases the cytoprotective effect of HT, possibly due to an increase in its effect on the production of peroxynitrite, since its antioxidant effect only increased at the lower concentration (Table 3). In EVOO, the concentration of DHPG is usually around 10% of that of HT [28]. If we extrapolate the results of LDH efflux with 100 µM HT and 10 µM DHPG, we observe that HT reduced this variable by 63%, DHPG by 9%, and the combination of both by 70%. A synergistic effect between both polyphenols was demonstrated to occur at the level of platelet function and lipid peroxidation in vitro [29].

A stimulating effect on the cytoprotective and antioxidant actions of HT was observed for OLC, although OLC did not modify the effect of HT on the production of peroxynitrite. It was demonstrated that OLC exerted an inhibitory effect on cyclooxygenase (COX) types 1 and 2 [30], which could indirectly decrease the production of lipid peroxides and promote the cytoprotective effect of HT; some nonsteroidal anti-inflammatory drugs have similar effects, especially those that inhibit both types of cyclooxygenases or, to a greater extent, COX-2 [31].

Through pharmacological analysis, the results of this study show which polyphenols in EVOO (DHPG and OLC) increase the cytoprotective effect of HT, which has not been previously demonstrated. However, this study has a fundamental limitation regarding the concentrations used. Our intention was to assess the existence of these interactions, but it is necessary to carry out the same experiments using the concentrations at which these polyphenols are found in brain tissue after the oral administration of an EVOO with a high polyphenol content.

## 5. Conclusions

This study shows that the EVOO polyphenols DHPG and OLC improve the protective effects of hydroxytyrosol in an in vitro model of the hypoxia-reoxygenation of rat brain slices, mainly due to a possibly synergistic effect at the level of its antioxidant action. These results could explain the greater cytoprotective effect of EVOO than of the polyphenols alone. Likewise, they could form the basis for the development of a functionalized oil with higher contents of DHPG and OLC in order to increase the cytoprotective effect of EVOO.

## Figures and Tables

**Figure 1 brainsci-11-01133-f001:**
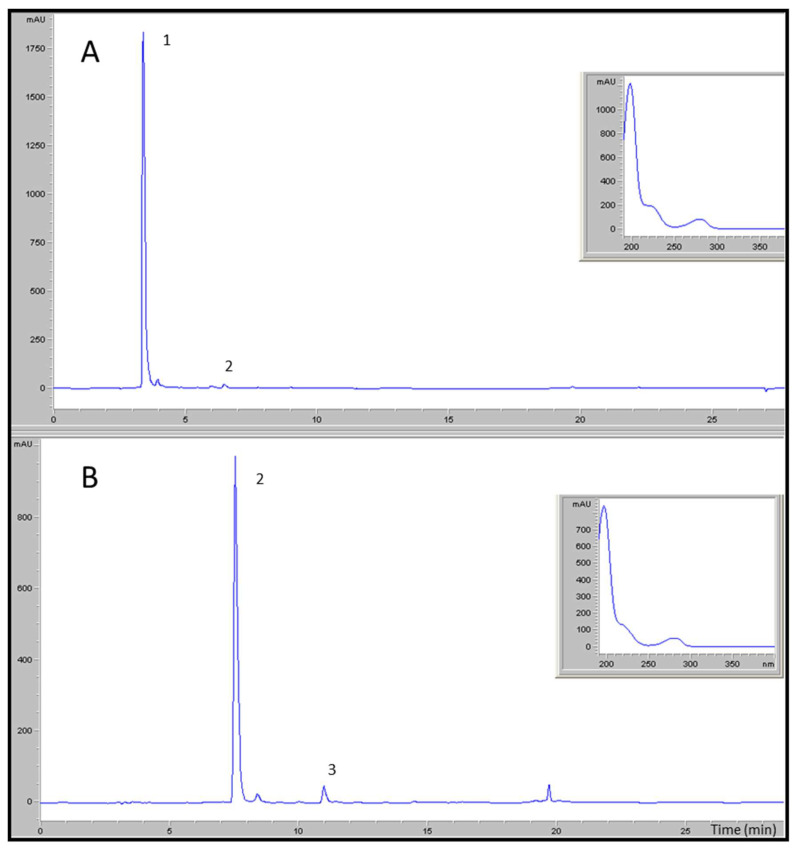
Chromatographic profiles at 280 nm of the purified 3,4-dihydroxyphenylglycol (**A**) and purified hydroxytyrosol (**B**) obtained from the HPLC–DAD and their UV–visible absorption spectra. The main compounds detected were (1) 3,4-dihydroxyphenylglycol, (2) hydroxytyrosol at 99.6%, and (3) tyrosol.

**Figure 2 brainsci-11-01133-f002:**
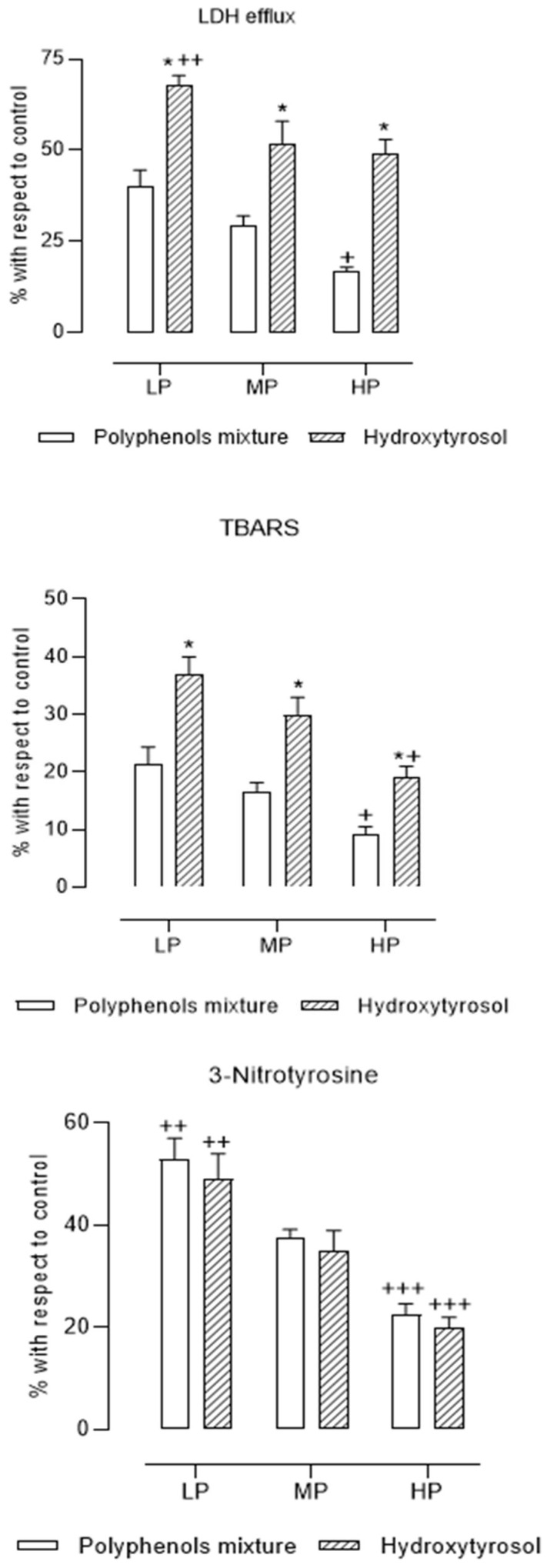
Mean values (mean ± SEM) of the percentages with respect to control samples of lactate dehydrogenase efflux (LDH efflux), thiobarbituric acid reactive substances (TBARS), and 3-nitrotyrosine in rat brain slices incubated with hydroxytyrosol, tyrosol, 3′,4′-dihydroxyphenylglycol, and oleocanthal in the same proportions of extra virgin olive oils considered to have low (LP), medium (MP), and high (HP) polyphenol contents (see Table 1) (N = 6 different experiments). * *p* < 0.0001 with respect to polyphenol mixture, + *p* < 0.05 with respect to LP and MP, ++ *p* < 0.05 with respect to MP and HP, and +++ *p* < 0.05 with respect to MP.

**Figure 3 brainsci-11-01133-f003:**
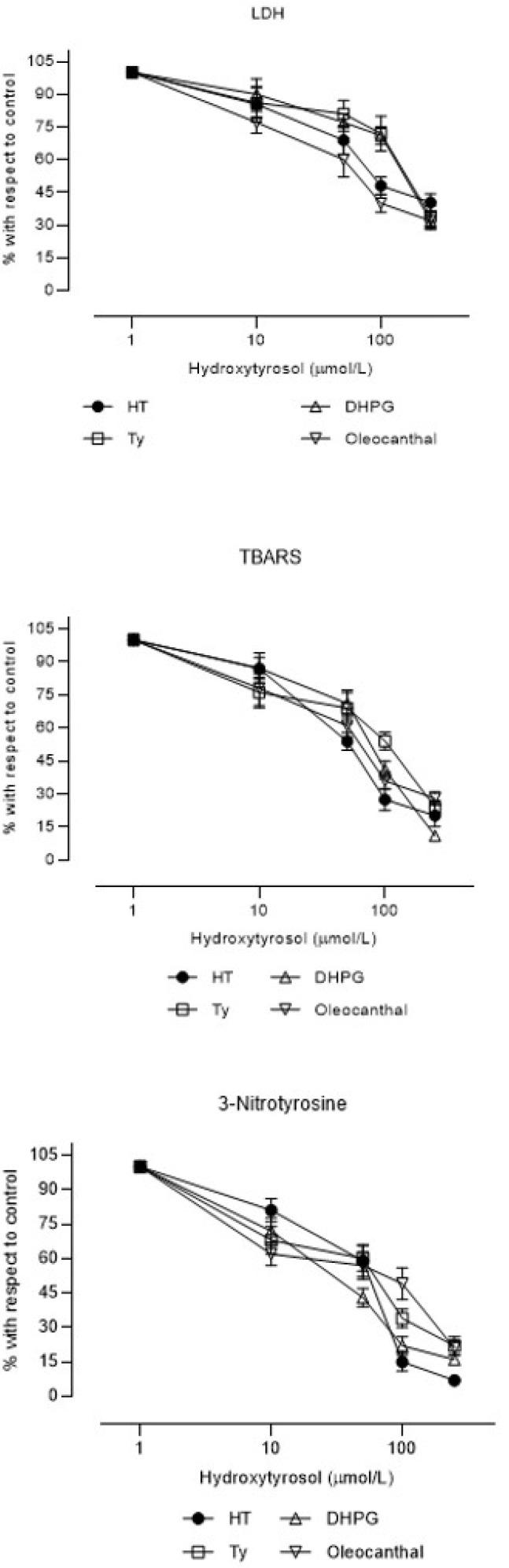
Concentration–effect curves (mean ± SEM) of the percentages with respect to control samples of lactate dehydrogenase efflux (LDH efflux), thiobarbituric acid reactive substances (TBARS), and 3-nitrotyrosine in rat brain slices incubated with hydroxytyrosol, tyrosol, 3′,4′-dihydroxyphenylglycol, and oleocanthal (N = 6 different experiments). All the compounds were incubated at 1, 10, 50, 100, and 250 µM.

**Figure 4 brainsci-11-01133-f004:**
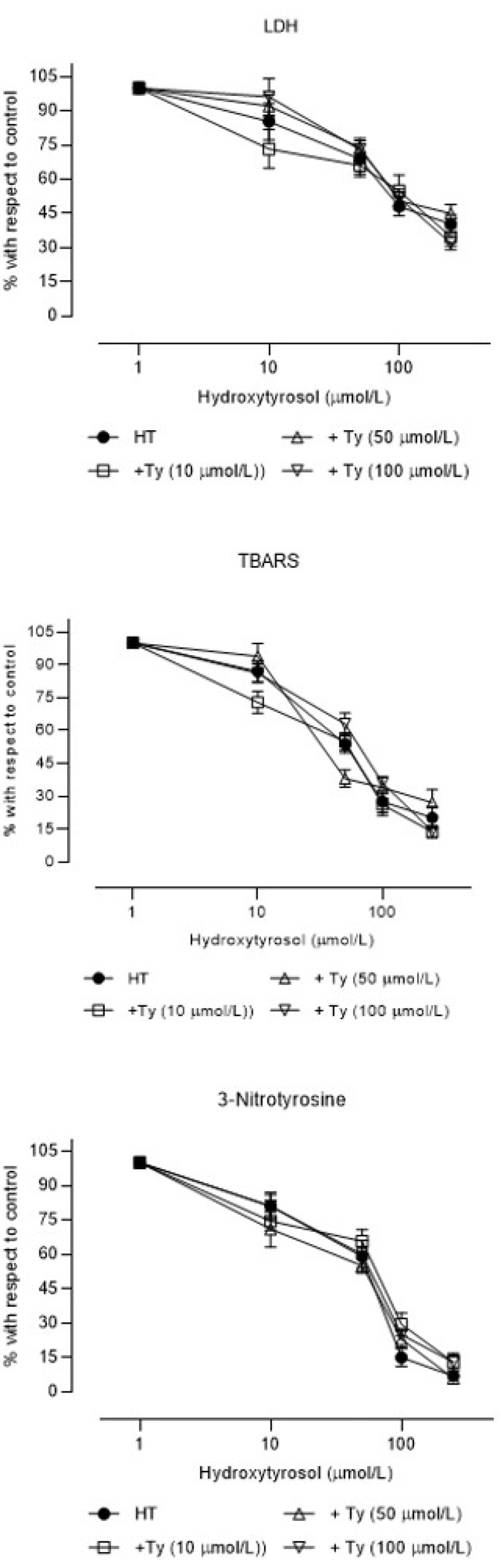
Concentration–effect curves (mean ± SEM) of the percentages with respect to control samples of lactate dehydrogenase efflux (LDH efflux), thiobarbituric acid reactive substances (TBARS), and 3-nitrotyrosine in rat brain slices incubated with hydroxytyrosol (HT) alone or in the presence of 10, 50, or 100 µM tyrosol (N = 6 different experiments).

**Figure 5 brainsci-11-01133-f005:**
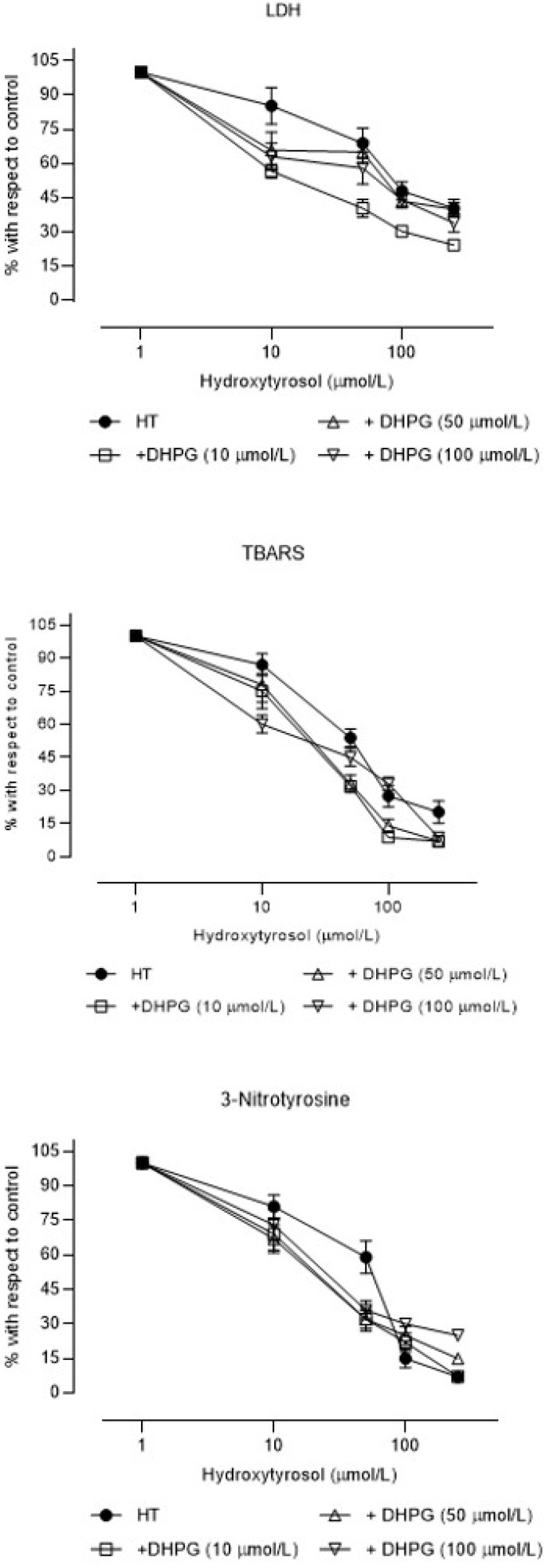
Concentration–effect curves (mean ± SEM) of the percentages with respect to control samples of lactate dehydrogenase efflux (LDH efflux), thiobarbituric acid reactive substances (TBARS), and 3-nitrotyrosine in rat brain slices incubated with hydroxytyrosol (HT) alone or in the presence of 10, 50, or 100 µM 3′,4′-dihydroxyphenylglycol (DHPG) (N = 6 different experiments).

**Figure 6 brainsci-11-01133-f006:**
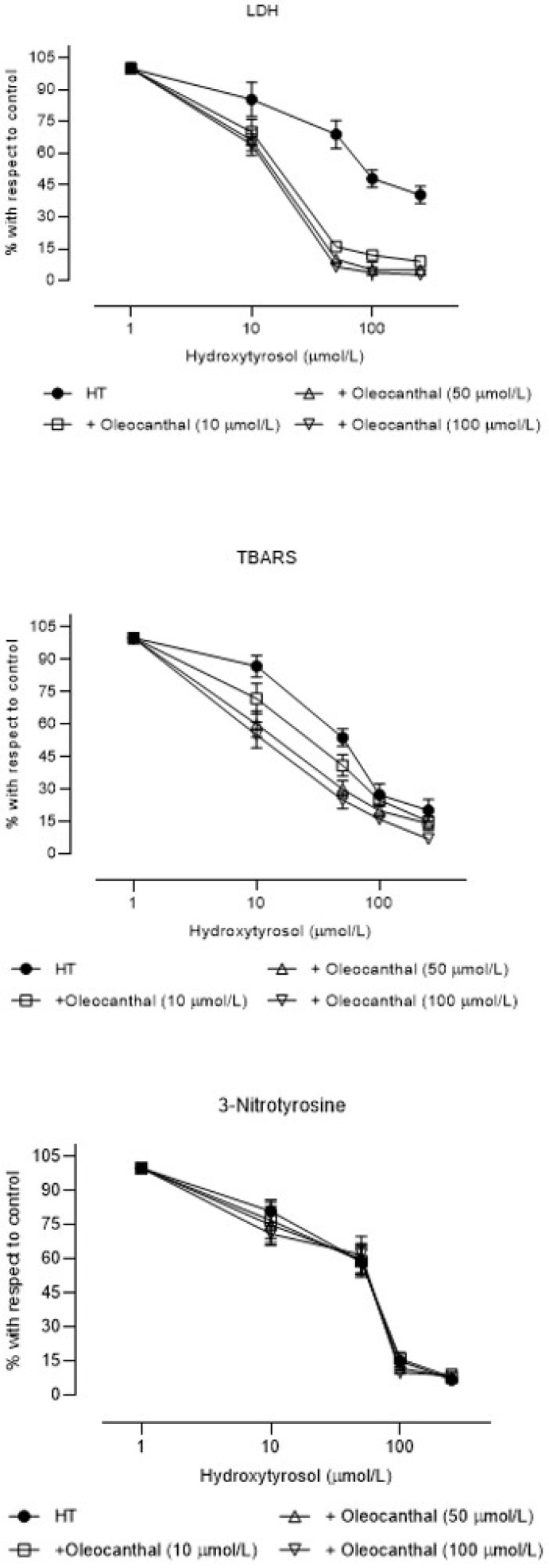
Concentration–effect curves (mean ± SEM) of the percentages with respect to control samples of lactate dehydrogenase efflux (LDH efflux), thiobarbituric acid reactive substances (TBARS), and 3-nitrotyrosine in rat brain slices incubated with hydroxytyrosol (HT) alone or in presence of 10, 50, or 100 µM oleocanthal (N = 6 different experiments).

**Table 1 brainsci-11-01133-t001:** Concentration ranges (mg/kg) of hydroxytyrosol (HT), tyrosol (Ty), 3′,4′-dihydroxyphenylglycol (DHPG), and oleocanthal (OLC) in extra virgin olive oils, considered to have low (LP), medium (MP), and high (HP) polyphenol contents.

-	HT	Ty	DHPG	OLC
EVOO—LP	-	-	-	-
Range	1.0–5.0	-	0.0–0.5	10–15
(incubated)	2.5	0.25	0.25	20
EVOO—MP	-	-	-	-
Range	10–50	0.5–1.1	1.0–5.0	100–200
(incubated)	30	0.8	0.3	150
EVOO—HP	-	-	-	-
Range	100–300	2.0–8.0	10–25	250–400
(incubated)	200	5.0	20	325

These data have been weighted from the references [13,14]. The concentration of each polyphenol that was chosen in the in vitro experiments is indicated as (incubated).

**Table 2 brainsci-11-01133-t002:** Mean values (mean ± SEM) of LDH efflux, thiobarbituric acid reactive substances (TBARS), and 3-nitrotyrosine (3-NTy) in rat brain slices subjected to a hypoxia-reoxygenation process (N = 6).

-	Pre-Hypoxia	Post-Reperfusion
LDH efflux (OD_492_–OD_600_ nm)	0.7 ± 0.1	3.2 ± 0.3 *
TBARS (nmol/mg protein)	0.8 ± 0.06	5.9 ± 0.6 *
3-NTy (nmol/mg tissue)	0.3 ± 0.02	2.7 ± 0.2 *

* *p* < 0.0001 with respect to pre-hypoxia values.

**Table 3 brainsci-11-01133-t003:** Mean values (mean ± SEM) of the concentrations that produced a 50% inhibition with respect to control samples (IC_50_, µM) of lactate dehydrogenase efflux (LDH), thiobarbituric acid reactive substances (TBARS), and 3-nitrotyrosine (3-NTy) in brain tissue after the hypoxia-reoxygenation model (N = 6 different experiments per group).

Compound	LDH	TBARS	3-NTy
Hydroxytyrosol (HT)	81.1 ± 4.7 *****	57.1 ± 5.7 **	64.4 ± 5.8
Tyrosol (Ty)	425 ± 25.1	154 ± 13.3 ***	68.9 ± 4.4
DHPG	314 ± 22.5	84.7 ± 6.3	40.5 ±3.9 *
Oleocanthal (OLC)	78.9 ± 5.8 *****	82.0 ± 6.3	75.3 ± 6.4
-	-	-	-
HT + Ty (10 µM)	185 ± 12.6 ****	51.3 ± 6.5	71.5 ± 5.7
HT + Ty (50 µM)	227 ± 8.2 ****	41.7 ± 3.9	57.6 ± 4.2
HT + Ty (100 µM)	125 ± 11.2 ****	91.4 ± 5.0 ****	84.2 ± 8.4
-	-	-	-
HT + DHPG (10 µM)	28.3 ± 3.4 ****	56.0 ± 5.2	27.7 ± 4.8 ****
HT + DHPG (50 µM)	85.9 ± 8.3	30.1 ± 3.8 ****	29.4 ± 4.7 ****
HT + DHPG (100 µM)	84.3 ± 8.2	49.8 ± 5.6 ****	32.9 ± 4.5 ****
-	-	-	-
HT + OLC (10 µM)	19.1 ± 2.6 ****	34.8 ± 2.9 ****	67.2 ± 7.1
HT + OLC (50 µM)	21.2 ± 1.8 ****	26.0 ± 2.8 ****	65.3 ± 6.6
HT + OLC (100 µM)	18.3 ± 1.1 ****	21.2 ± 2.7 ****	68.0 ± 5.7

* *p* < 0.05 with respect to HT, Ty, and OLC. ** *p* < 0.01 with respect to Ty, DHPG, and OLC. *** *p* < 0.001 with respect to HY, DHPG, and OLC. **** *p* < 0.005 with respect to HT. ***** *p* < 0.0001 with respect to Ty and DHPG.

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
