# Peer review of "Extra Virgin Oil Polyphenols Improve the Protective Effects of Hydroxytyrosol in an In Vitro Model of Hypoxia-Reoxygenation of Rat Brain"

_brainsci, 2021, doi:10.3390/brainsci11091133_

Round 1

Reviewer 1 Report

The manuscript completed by José Pedro De La Cruza et al. reported an in vitro study showing that EVOO polyphenols, DHPG and OLC, increase the neuroprotective effect of HT, mainly due 30 to a possible synergistic effect at the level of its antioxidant action.

Although this study is very interesting and kind of clinically relevant for brain health, however, the current study is suffering some flaws and major concerns in the experiment design and result, which greatly harm their conclusion.

  1. First of all, although the oxygen-glucose-deprivation was performed on the brain slice has been accepted as a model to evaluate the neuroprotective effect of certain compounds, however, histological evidence is required to prove neuron death or gliosis. Therefore, we strongly suggest the author provide the histological staining to confirm the neuronal loss that occurred during oxygen/glucose deprivation.

  1. Secondly, as figure 1 already demonstrated the HT exerts the maximal anti-oxidative effect at the dose of 200μM, the reviewer is concerning the necessaries the HT test at the lower doses in figure 2. Additionally, the authors did not state why the HT at 100 but not 200 μM was chosen in the Figure 2 test.

  1. For the data presented in Table 3, the author should explain why the HT combined with Ty moderate dose even worsen the oxidative stress induced by oxygen/glucose deprivation, compared to HT+Ty low dose.

In addition, the reviewer strongly suggested these authors same symbol (like the star) presented the comparison within or among groups. Generally, the number of stars indicates the degree of statistical significance, the more the stronger.

  1. For increasing the transparency of the methodology of the article, the authors is required to provide detailed information about the chemical used in the experiment.

1) Like the catalog information of chemical reagent and immunoassay or colorimetric kits mentioned in the Method Section.

2)  Since the Hydroxytyrosol (HT) and 3’,4’-dihidroxifenilglicol (DHPG) used in the article are not commercial chemicals, the author are strongly required to provide the chromatogram of HT and DHPG  analyzed in the HPLC experiment.

3)  For the 2.5.1. the section lacks the product and related protocol information.

  1. The current article contains incomplete statistical information. Like, in table 2, the authors used the unpaired t-test, at least the t value should be presented, which is followed by the p-value. Similar issues have been found in Figure 1 and Table 3, as the authors claimed that the One-Way ANOVA was utilized, both F and p values should be reported in the result or legend.

Author Response

  1. First of all, although the oxygen-glucose-deprivation was performed on the brain slice has been accepted as a model to evaluate the neuroprotective effect of certain compounds, however, histological evidence is required to prove neuron death or gliosis. Therefore, we strongly suggest the author provide the histological staining to confirm the neuronal loss that occurred during oxygen/glucose deprivation.

Response: The title of the manuscript has been changed according to the suggestion of the reviewer #2. Also we have taken into account the “cytoprotective” instead “neuroprotective” effect of the results obtained in this study.

  1. Secondly, as figure 1 already demonstrated the HT exerts the maximal anti-oxidative effect at the dose of 200μM,  . Additionally, the authors did not state why the HT at 100 but not 200 μM was chosen in the Figure 2 test.

Response: The concentrations of HT incubated in each experiment were: 1, 10, 50, 100 and 250 µM. In previous experiments (Food Chem 2012, 134, 2176-2183) we demonstrated that concentrations of HT higher than 500 µM exerted a progressive decrease in most of the variables evaluated in rat brain slices. For that reason, we did not incubate concentrations over 250 µM

  1. For the data presented in Table 3, the author should explain why the HT combined with Ty moderate dose even worsen the oxidative stress induced by oxygen/glucose deprivation, compared to HT+Ty low dose.

Response: The results shown in Table 3 indicate that Ty slows the antioxidant effect of HT when incubated at high concentrations, not interfering with this effect at medium or low concentrations. This occurs with drugs called partial agonists (AgP), which act by the same mechanism of action as the agonist (Ag) but with much less potency. In these cases, when high concentrations of AgP are incubated together with Ag, it behaves as an antagonist. The same hypothesis could be postulated when analysing the IC50 results for LDH efflux. This explanation is expressed in the discussion (fourth paragraph). In any case, the Ty plus HT association does not worsen the levels of lipid peroxidation induced by the hypoxia-reoxygenation model, but rather the antioxidant effect of HT is decreased.

In addition, the reviewer strongly suggested these authors same symbol (like the star) presented the comparison within or among groups. Generally, the number of stars indicates the degree of statistical significance, the more the stronger.

 Response: All symbols in Table 3 have been replaced by the star (*). Moreover, the number of stars has been ordered according to the order from lowest to highest statistical significance.

  1. For increasing the transparency of the methodology of the article, the authors is required to provide detailed information about the chemical used in the experiment.

1) Like the catalog information of chemical reagent and immunoassay or colorimetric kits mentioned in the Method Section.

Response: A more detailed description of these determinations has been included in the text.

2)  Since the Hydroxytyrosol (HT) and 3’,4’-dihidroxifenilglicol (DHPG) used in the article are not commercial chemicals, the author are strongly required to provide the chromatogram of HT and DHPG  analyzed in the HPLC experiment.

Response: These chromatograms have been included as Figure 1. All other figures have been renumbered.

3)  For the 2.5.1. the section lacks the product and related protocol information.

Response: A more detailed description of these determinations has been included in the text.

  1. The current article contains incomplete statistical information. Like, in table 2, the authors used the unpaired t-test, at least the t value should be presented, which is followed by the p-value. Similar issues have been found in Figure 1 and Table 3, as the authors claimed that the One-Way ANOVA was utilized, both F and p values should be reported in the result or legend.

Response: We have indicated in each group of results the differences that reached statistical significance, leaving no symbols or p data for the differences and comparisons that were not statistically significant. We also decided to include those differences that were directly related to the objectives of the study. However, we have included the rest of the comparisons between groups (new Fig 2) and rearranged the significance in Table 3.

Reviewer 2 Report

The data presented by De la Cruz et al. show how some EVOO polyphenols act synergically with hydroxytyrosol (HT) in protecting brain from injury occurring in an in vitro model of hypoxia-reoxygenation. The authors have compared the effect of HT alone with HT in combination with tyrosol (Ty), 3', 4'-dihydroxyphenylglycol (DHPG) and oleocanthal (OLC) on LDH efflux, lipid peroxidation and peroxynitrite production.

The data are convincing, however some additional experiments and text revision must be performed before publication

Here some specific comments:

1) Modulation of LDH efflux alone is not sufficient to affirm the presence of a neuroprotective effect. It should be better for the specific experimental system used (hypoxia-reoxygenationin brain slice) to refer to brain injury. So please change this issue everywhere in the text and in the title. For example, the title could be: Extra virgin oil polyphenols improve the protective effects of hydroxityrosol in vitro model of hypoxia-reoxygenation of rat brain

2) It is necessary extensive revision of English language and style. Some sentences are rather obscure and should be rephrased. Here some examples: “However, it is less than that of EVOO, for that reason it is postulated that there could be an interaction between several of the polyphenols of EVOO” (lane 17). And: In all cases, the neuroprotective and antioxidant effect of the combination (with the other EVOO polyphenols) was greater than that of HT (alone) (lane 23).

3) Why the choice of Ty, DHPG and OLC? The rationale has not been explained. A preliminary experiment with the identification of active ingredients of EVOO having the protective effects using the alternate combinatory treatment (with one or more polyphenols) for example only on one parameter, such as LDH efflux could demonstrate the rationale for  the choice. At this regard, I suggest to give information about HT in combination with oleuropein aglycone (OA), that is the most abundant phenolic constituent of EVOO with a key role in counteracting Aβ42 toxicity in the rat brain (Neurosci Lett. 2014 Jan 13;558:67-72).

3) The sentence at lane 21 in the abstract. It should be better specified the reason of the measurement performed. For example: LDH efflux, lipid peroxidation and peroxynitrite production were determined as a measure of cell death, oxidative stress and nitrosative stress, respectively. At this regard, I suggest to add information on the effects on the energy source known to be also impaired during hypoxia-reoxygenation. Indeed, interventions that reduce the cellular damage in hypoxia-reoxygenation injury may target: (1) the metabolism and energy resources, (2) the oxidative stress pathways and antioxidant responses or (3) the proteasome and proteolytic activity. The authors give information on point 2. I suggest to give also information on the point 1 by measuring the ATP level as additional information of the polyphenols effect that could target dysfunctional mitochondria and improve the brain ATP level. In addition, at this regards I would leave the “main objective” and the “secondary objective” of this study (lanes 65-68). The secondary objective is as important as the “main”. Actually, without assessing the mechanistic aspect of the polyphenols effects the data would be unpublishable.

4) Lane 126: “All compounds were incubated from the beginning of the experiments” what it means, before hypoxia as preventive effects? For how long? Please give more experimental details.

5) Figure 1. Compare and give significance in the difference among  LP, MP and HP incubation.

6) Figure 2. The concentration of tyrosol, 3',4'-dihydroxyphenylglycol and oleocanthal incubated with HT should be reported in the legend

Author Response

1) Modulation of LDH efflux alone is not sufficient to affirm the presence of a neuroprotective effect. It should be better for the specific experimental system used (hypoxia-reoxygenationin brain slice) to refer to brain injury. So please change this issue everywhere in the text and in the title. For example, the title could be: Extra virgin oil polyphenols improve the protective effects of hydroxityrosol in vitro model of hypoxia-reoxygenation of rat brain

Response: The title of the manuscript has been changed according to the suggestion of the reviewer. Also we have taken into account the “cytoprotective” instead “neuroprotective” effect of the results obtained in this study.

2) It is necessary extensive revision of English language and style. Some sentences are rather obscure and should be rephrased. Here some examples: “However, it is less than that of EVOO, for that reason it is postulated that there could be an interaction between several of the polyphenols of EVOO” (lane 17). And: In all cases, the neuroprotective and antioxidant effect of the combination (with the other EVOO polyphenols) was greater than that of HT (alone) (lane 23).

Response: This revised version of our manuscript has been revised by MDPI English Editing Service

3) Why the choice of Ty, DHPG and OLC? The rationale has not been explained. A preliminary experiment with the identification of active ingredients of EVOO having the protective effects using the alternate combinatory treatment (with one or more polyphenols) for example only on one parameter, such as LDH efflux could demonstrate the rationale for  the choice. At this regard, I suggest to give information about HT in combination with oleuropein aglycone (OA), that is the most abundant phenolic constituent of EVOO with a key role in counteracting Aβ42 toxicity in the rat brain (Neurosci Lett. 2014 Jan 13;558:67-72).

Response: We chose these four EVOO compounds because they are recognized as the simple polyphenols with more biological potency than those in EVOO. Some others, such as oleuropein, act through their hydroxytyrosol content, mainly by releasing it in the stomach when oleuropein is ingested orally. In this way, hydroxytyrosol is considered, either in a simple way or released from oleuropein, the polyphenol with the highest biological activity of EVOO. That is why we chose these simple polyphenols, as well as the reason for studying the effect of Ty, DHPG and OLC on the effect of HT. In the revised version of our article we have added a sentence to explain this concept (Introduction, paragraph 4).

 4) The sentence at lane 21 in the abstract. It should be better specified the reason of the measurement performed. For example: LDH efflux, lipid peroxidation and peroxynitrite production were determined as a measure of cell death, oxidative stress and nitrosative stress, respectively. At this regard, I suggest to add information on the effects on the energy source known to be also impaired during hypoxia-reoxygenation. Indeed, interventions that reduce the cellular damage in hypoxia-reoxygenation injury may target: (1) the metabolism and energy resources, (2) the oxidative stress pathways and antioxidant responses or (3) the proteasome and proteolytic activity. The authors give information on point 2. I suggest to give also information on the point 1 by measuring the ATP level as additional information of the polyphenols effect that could target dysfunctional mitochondria and improve the brain ATP level. In addition, at this regards I would leave the “main objective” and the “secondary objective” of this study (lanes 65-68). The secondary objective is as important as the “main”. Actually, without assessing the mechanistic aspect of the polyphenols effects the data would be unpublishable.

Response: Thank you very much for the suggestions. Obviously, the exploration of the possible effect of these polyphenols on the metabolic-energy state of brain cells and on the functionalism of mitochondria in the experimental model is a very interesting proposal. In this study we wanted to find out whether or not there were interactions between them, for which we chose a mechanism accepted as fundamental in tissue ischemia-reperfusion processes (oxidative and nitrosative stress). Once the results of this interaction have been obtained, in future studies we will be able to address the exact mechanism by which these polyphenols interact in the experimental model used.

The proposed sentence has been added in abstracts.

We have eliminated the difference between main and secondary objectives, leaving only one type of objective.

5) Lane 126: “All compounds were incubated from the beginning of the experiments” what it means, before hypoxia as preventive effects? For how long? Please give more experimental details.

Response: All compounds were incubated from the beginning of the experiment (Period A) and kept until the end of the reoxygenation period (Period C). This sentence has been added in section 2.4.

6) Figure 1. Compare and give significance in the difference among  LP, MP and HP incubation.

Response: This analysis has been included in the new Fig 2 (Fig 1 in the first version of the manuscript)

7) Figure 2. The concentration of tyrosol, 3',4'-dihydroxyphenylglycol and oleocanthal incubated with HT should be reported in the legend.

Response: This figure represents the effect of the compounds studied separately, they were not incubated associated in any case. However, the incubated concentrations of these compounds have been added in the legend to Figure 2.

Round 2

Reviewer 1 Report

The reviewer's concerns have been well addressed.